# Blood Lactate and Maximal Lactate Accumulation Rate at Three Sprint Swimming Distances in Highly Trained and Elite Swimmers

**DOI:** 10.3390/sports11040087

**Published:** 2023-04-19

**Authors:** Maria Mavroudi, Athanasios Kabasakalis, Anatoli Petridou, Vassilis Mougios

**Affiliations:** Laboratory of Evaluation of Human Biological Performance, School of Physical Education and Sport Science at Thessaloniki, Aristotle University of Thessaloniki, 541 24 Thessaloniki, Greece; mmavroudi@phed.auth.gr (M.M.); kabasakalis@phed.auth.gr (A.K.); apet@phed.auth.gr (A.P.)

**Keywords:** anaerobic lactic power, sampling time, swimming speed, training

## Abstract

We examined the blood lactate response, in terms of the maximal post-exercise concentration (La_max_), time to reach La_max_, and maximal lactate accumulation rate (VLa_max_), to swimming sprints of 25, 35, and 50 m. A total of 14 highly trained and elite swimmers (8 male and 6 female), aged 14–32, completed the 3 sprints in their specialization stroke with 30 min of passive rest in between. The blood lactate was measured right before and continually (every minute) after each sprint to detect the La_max_. The VLa_max_, a potential index of anaerobic lactic power, was calculated. The blood lactate concentration, swimming speed, and VLa_max_ differed between the sprints (*p* < 0.001). The La_max_ was highest after 50 m (13.8 ± 2.6 mmol·L^–1^, mean ± SD throughout), while the swimming speed and VLa_max_ were highest at 25 m (2.16 ± 0.25 m·s^–1^ and 0.75 ± 0.18 mmol·L^–1^·s^–1^). The lactate peaked approximately 2 min after all the sprints. The VLa_max_ in each sprint correlated positively with the speed and with each other. In conclusion, the correlation of the swimming speed with the VLa_max_ suggests that the VLa_max_ is an index of anaerobic lactic power and that it is possible to improve performance by augmenting the VLa_max_ through appropriate training. To accurately measure the La_max_ and, hence, the VLa_max_, we recommend starting blood sampling one minute after exercise.

## 1. Introduction

Swimming is one of the most popular competitive sports worldwide, requiring a combination of physical abilities [1] such as strength, speed, power, and endurance to achieve peak performance, due to the large range of distances and intensities in swimming events [2,3]. These abilities are supported by a combination of the ATP–phosphocreatine, lactate, and aerobic energy systems [2,4,5]. The determination of the capacity and power of each of these energy systems is an integral part of swimming training. Although an assessment of aerobic power is rather straightforward (through maximal oxygen uptake, VO_2_max), the estimation of anaerobic power (both lactic and alactic) requires complex procedures, such as a muscle biopsy, the measurement of the maximal blood lactate concentration, and the calculation of the maximal blood lactate accumulation rate (VLa_max_) [6,7,8].

Examining the blood lactate response is a common way of determining anaerobic power in swimming. Anaerobic carbohydrate catabolism dominates the 50 and 100 m events, lasting approximately 20 s to 1 min [4,5]. Research has shown that the highest muscle and blood lactate accumulation in elite swimmers occurs in the 100 and 200 m events, with the blood lactate reaching 14 mmol·L^–1^ on average [9]. The blood lactate in 50 m swimming sprints ranges from approximately 7 to 12 mmol·L^–1^ [3,9,10,11]. Studies on repeated swimming distances below 50 m have reported blood lactate concentrations ranging from as low as 4 mmol·L^–1^ to as high as 14 mmol·L^–1^ [12,13,14]. However, the research on single swimming sprints is limited and reports blood lactate concentrations of up to 22 mmol·L^–1^ after 15 m sprints [15].

In addition to the maximal post-exercise blood lactate concentration (La_max_), the time needed for blood lactate to peak after the end of maximal exercise (tLa_max_) has been a parameter of interest for researchers because knowing it is required for the accurate measurement of the La_max_. The La_max_ usually appears immediately or a few minutes post-exercise, thus necessitating repeated and frequent blood sampling for approximately 10 min if blood samples are to be analyzed later or, if lactate is measured on the spot using a portable device, until a value lower than the previous one is detected [4,5].

The VLa_max_ (the aforementioned potential index of anaerobic lactic power) is calculated as the difference between the La_max_ and the pre-exercise blood lactate concentration (La_pre_), divided by the difference between the exercise time (t_exer_) and a period at the onset of exercise during which it is assumed that increased lactate production has not yet begun (t_alac_) [6,16,17]. Thus, the mathematical expression of VLa_max_ is
VLa_max_ = (La_max_ − La_pre_)/(t_exer_ − t_alac_)(1)

Of the terms in Equation (1), all but t_alac_ are determined experimentally. t_alac_ can be estimated using several approaches. Heck and coworkers [6] proposed values of 3 s for an exercise duration of up to 10 s, 4 s for an exercise duration of 20 s, and 8 s for an exercise duration of 50 to 60 s. Because different estimates of t_alac_ may have a considerable impact on the VLa_max_, an accurate determination of this parameter is important [18].

The VLa_max_ has been determined using short maximal exercise tests in cycling [16,19,20,21], hand cycling [21], running [8,20], and swimming [22,23]. It is believed that athletes with higher VLa_max_ values have increased lactate synthesis at both low and high power outputs [17]. Thus, Wackerhage and coworkers [17] stress the importance of a high VLa_max_ for events where power output reaches very high values for periods ranging from a few seconds to just over a minute; they have also shown an inverse relationship between the VLa_max_ and maximal lactate steady state in elite sprinters. Therefore, the VLa_max_ can be used to estimate anaerobic power, determine individual training needs, and design appropriate exercises to improve performance in many sports [8]. However, the same athletes may display different VLa_max_ values in different sports [20].

Based on the limited available scientific data regarding the blood lactate response (in terms of the La_max_, tLa_max_, and VLa_max_) in sprint swimming and particularly distances up to 50 m, the present research aimed at investigating the blood lactate response of highly trained and elite swimmers to maximal efforts of 25, 35, and 50 m. According to Heck and coworkers [6], the most appropriate exercise time for calculating the VLa_max_ is 10–15 s, which corresponds to 25 m sprints in swimming; 50 m is the shortest racing distance, while 35 m is an intermediate distance. This work has potential utility at both the theoretical and practical levels. In terms of theory, we provide novel data on the involvement of the lactate system in energy provision during swimming sprints. In terms of practice, we provide coaches and exercise scientists with information on how to better assess anaerobic lactic power and implement training and competition strategies to maximize performance.

## 2. Materials and Methods

### 2.1. Participants

A total of 14 (8 male and 6 female) swimmers with a mean of 742 FINA points (range, 628–873), aged 14–32 years, and specialized in sprint events (50 and 100 m), participated in the study. The sample size was dictated by the number of highly trained (tier 3, national level) and elite (tier 4, international level) swimmers [24] who we had access to and who consented to participate, along with their coaches and (in the case of underage individuals) their parents. Sample size was not calculated on the basis of expected statistical power. Nevertheless, as will be shown, this study was sufficiently powered (power > 0.8).

### 2.2. Ethics

Athletes and the parents of the underage ones were informed orally and in writing about the details of the experimental procedure, after which they provided written consent. Procedures were in accordance with the Declaration of Helsinki, and the study design was approved by the Research Ethics Committee of the School of Physical Education and Sport Science at Thessaloniki, Aristotle University of Thessaloniki (approval number 89/2021).

### 2.3. Anthropometric Measures

Body mass was measured to the nearest 0.1 kg by an electronic balance (Seca, Hamburg, Germany). Height was measured to the nearest 0.01 m by a stadiometer fixed to the balance, and body mass index (BMI) was calculated from the 2 measures.

### 2.4. Exercise Protocol

The experimental procedure was performed in the afternoon (4–6 pm), in a 50 m indoor swimming pool, at ambient air temperature of 29 °C and water temperature of 26.5 °C. Participants were asked to avoid strenuous training during the previous 48 h. After a light 10 min warm-up at self-selected pace, each athlete completed 3 swimming sprints of 25, 35, and 50 m in that order (that is, from lowest to highest load) and in the swimming stroke each one was specialized in. After completing each distance, they rested for 30 min. All sprints were performed with competitive starts, and swimmers were instructed to keep their fastest possible pace throughout the tests. Time to complete each sprint (t_exer_) was recorded by an experienced coach and member of the research group (AK) using a digital stopwatch. For the 25 and 35 m sprints, time was determined from the start signal until the swimmer’s hand crossed the 25 or 35 m mark, respectively. Swimming speed was calculated from all these times.

### 2.5. Blood Lactate Measurement

Lactate was measured in capillary blood from a fingertip right before (La_pre_) and continually after each sprint, starting at 30 s and every minute thereafter until a value that was lower than the previous one (declared the La_max_) was measured, using a portable analyzer (Lactate Scout 4, EΚF Diagnostics, Cardiff, UK). tLa_max_ after each sprint was estimated by plotting the concentration against time in an MS Excel worksheet.

### 2.6. Calculation of VLa_max_

VLa_max_ in each distance was calculated according to Equation (1), in which La_max_, La_pre_, and t_exer_ were as described above. t_alac_ was estimated based on Heck and coworkers [6]. Because their estimates include exercise durations of only 10, 20, and 50–60 s; and because we had exercise durations ranging from 10.14 to 31.88 m, we fine-tuned the estimates of t_alac_ as follows:If exercise lasted 10.01 to 14.99 s, t_alac_ was 3.5 s.If exercise lasted 15.00 to 19.99 s, t_alac_ was 4 s.If exercise lasted 20.00 to 24.99 s, t_alac_ was 4.5 s.If exercise lasted 25.00 to 29.99 s, t_alac_ was 5 s.If exercise lasted 30.00 to 34.99 s, t_alac_ was 5.5 s.

To examine the robustness of these estimates, we also calculated VLa_max_ based on 2 extreme scenarios, 1 assuming t_alac_ to be null in all cases (thus assuming that increased lactate production begins at the very onset of maximal exercise) and 1 assuming t_alac_ to be 1.5 times the values listed above. (We consider this an upper extreme of t_alac_ because it does not exceed half the exercise duration in any of these cases, as would be expected of exercises relying primarily on the lactate system [5].)

### 2.7. Statistical Analysis

Normality of data distribution was tested through the Shapiro–Wilk test and was found not to differ significantly from normal for any parameter (*p* ranging from 0.142 to 0.999). Descriptive data are reported as mean (SD). For the statistical analysis of swimming speed, tLa_max_, and VLa_max_, analysis of variance (ANOVA) with repeated measures on distance was performed. For the analysis of blood lactate, 2-way ANOVA, that is, distance with 3 levels (25, 35, and 50 m) by time with 2 levels (La_pre_ and La_max_) with repeated measures on both factors was performed. Effect sizes for significant main effects and interactions were determined as partial η^2^ and were classified as small (0.01–0.058), medium (0.059–0.137), or large (>0.137) according to Cohen [25]. The observed statistical power was also recorded. Significant outcomes were followed up by simple main effect analysis with Sidak adjustment for multiple comparisons. Correlations between parameters were investigated by Pearson’s correlation analysis. The significance level for all tests was set at α = 0.05. The SPSS version 27.0 (SPSS, Chicago, IL, USA) was used for all analyses.

## 3. Results

### 3.1. Characteristics of Participants

The demographic and anthropometric characteristics of the swimmers are presented in Table 1. The specialized swimming stroke and FINA points of each swimmer are shown in Table 2.

### 3.2. Swimming Performance

The swimming times and speeds at the 25, 35, and 50 m sprints of each swimmer are presented in Table 3. The swimming speed differed significantly between distances (*p* < 0.001, η^2^ = 0.955, power > 0.999), with all pairwise comparisons being significant (*p* < 0.001); the speed was highest at 25 m and lowest at 50 m.

### 3.3. Blood Lactate Response

Table 4 presents the blood lactate response to the sprints. The significant main effects of time (*p* < 0.001, η^2^ = 0.964, power > 0.999) and distance (*p* < 0.001, η^2^ = 0.925, power > 0.999), as well as a significant interaction of the time and distance on the blood lactate concentration was found (*p* < 0.001, η^2^ = 0.703, power > 0.999). All pairwise comparisons (that is, according to the main effect and interaction) revealed significant differences (*p* < 0.001). The La_pre_, La_max_, and change in the lactate concentration with exercise increased bout after bout.

Regarding the tLa_max_ (Table 4), there was no significant difference between the sprints. The tLa_max_ averaged 2.2 min and ranged from 1.6 to 4.7 min. A significant correlation between the tLa_max_ in the 35 and 50 m sprints was found (*r* = 0.560, *p* = 0.037).

The VLa_max_ differed significantly between the sprints (*p* < 0.001, η^2^ = 0.695, power > 0.999) (Table 4), being highest at the 25 m sprint and lowest at the 50 m sprint. Significant differences were located between 25 and 35 m, as well as between 25 and 50 m (both *p* < 0.001). The VLa_max_ for all the sprints was positively correlated with the speed at all of the sprints (that is, nine significant correlations). Of these, the correlation coefficients of the VLa_max_ with speed within the same sprint (that is, the most relevant correlations) were 0.541 (*p* = 0.046) at 25 m, 0.587 (*p* = 0.027) at 35 m, and 0.839 (*p* < 0.001) at 50 m. In addition, all 3 VLa_max_ were positively correlated with each other (with *r* ranging from 0.636, *p* = 0.014, to 0.821, *p* < 0.001).

When the VLa_max_ was calculated according to the two alternative scenarios described under Methods, we obtained the following results: With the “null” scenario, the VLa_max_ was 0.52 (0.11), 0.39 (0.12), and 0.40 (0.13) for 25, 35, and 50 m, respectively (*p* < 0.001, η^2^ = 0.394, power = 0.944). With the “1.5×” scenario, the VLa_max_ was 0.96 (0.26), 0.63 (0.21), and 0.55 (0.19), respectively (*p* < 0.001, η^2^ = 0.98, power > 0.999). With both scenarios, the VLa_max_ differed significantly between 25 and 35 m, as well as between 25 and 50 m (both *p* < 0.001) but not between 35 and 50 m. The VLa_max_ was positively correlated with the speed in each sprint (*p* < 0.05), except for 25 and 35 m in the “null” scenario. Finally, all 3 VLa_max_ were positively correlated with each other in the “1.5×” scenario (*p* < 0.01), but no correlation between the 3 VLa_max_ was significant in the “null” scenario.

## 4. Discussion

The aim of the present study was to examine the blood lactate response (in terms of the La_max_, tLa_max_, and VLa_max_) of highly trained and elite swimmers specialized in sprint events to the maximal efforts of 25, 35, and 50 m. Our main (and novel) findings are that the VLa_max_ was highest at 25 m, the VLa_max_ was correlated with the swimming speed, and the tLa_max_ was short and not different between bouts. It is worth noting that the statistical power was high in all cases.

The progressively higher increase in the blood lactate concentration after each sprint can be explained by the higher work required with increasing distance [26]. To the best of our knowledge, there are no La_max_ values after single swimming sprints of 25 or 35 m in the literature. Affonso and coworkers [15] have reported a mean La_max_ of 20.5 mmol·L^–1^ after a 15 m sprint with a block start in 3 world-class swimmers, a value that is considerably higher than the ones measured in the present study. This difference may be due to the different characteristics of the participants in the two studies. Regarding the La_max_ after the 50 m sprint, our values (mean 13.8 mmol·L^–1^) are higher than those reported by Invernizzi and coworkers [10], Vescovi and coworkers [9], Campos and coworkers [3], and Lisbôa and coworkers [11] (ranging between 9.1 and 12.2 mmol·L^–1^). Again, these differences may be attributed to different characteristics of the participants and different methods of lactate measurement. Another possible factor is the different training period that each athlete was in, which seems to affect the La_max_ [27,28].

The tLa_max_ found after all 3 sprints of the present study (about 2 min) was shorter than what is usually thought, that is, between 4 and 10 min [4,5,29]. The reason for this difference in the tLa_max_ is not clear. Possible explanations include exercise modality, muscle fiber type composition, and training background.

Apart from the maximal post-exercise blood lactate concentration and the time needed to reach it, the maximal rate of lactate production, estimated through the VLa_max_, is of great interest as a potential index of anaerobic lactic power. In the present study, we show that the VLa_max_ was highest after the shortest swimming distance (25 m, which lasted about 12 s) and lowest after the longest one (50 m, which lasted about 27 s). This agrees with Heck and coworkers [6], who suggested assessing the VLa_max_ in maximal efforts lasting about 10 s and justifies the choice of maximal efforts lasting 10 to 15 s to determine the highest VLa_max_ value by other researchers [8,16,20,21]. On the other hand, it should be noted that the correlation between the VLa_max_ and swimming speed was highest in the 50 m sprint (which lasted about 27 s), with a prediction percentage (resulting from *r*^2^) of 70%, as opposed to 34% in the 35 m sprint, and 29% in the 25 m sprint. This suggests that maximal bouts lasting longer than 15 s and still relying mainly on the lactate system, such as the 30 s Wingate test employed by Hommet and coworkers [19], may be more appropriate for predicting the performance from the VLa_max_.

The positive correlation of the VLa_max_ with the swimming speed in all three sprints, as opposed to the absence of any correlation of the La_max_ with speed, suggests that the VLa_max_ is a better index of anaerobic lactic power than the La_max_. The positive correlation between the VLa_max_ values at all three distances adds to the validity of this index.

It is remarkable that the order of the La_max_ in the 3 sprints (that is, lowest at 25 m and highest at 50 m) was the reverse of the order of the VLa_max_. These reverse trends are in line with the reverse trends in the speed and distance in swimming events, in which speed decreases with increasing distance. The underlying reason for both reverse trends is that the fastest energy sources are also the smallest, whereas the largest energy sources are also the slowest.

In the present study, we have proposed estimates of t_alac_ per 5 s increments of t_exer_, which are more fine-tuned compared to those proposed by Heck and coworkers [6]. We further examined the robustness of the estimates by varying them through two extreme scenarios, as described under Methods. The “null” scenario (the one assuming t_alac_ to be 0 in all cases) did not perform well, because the VLa_max_ calculated on this basis did not correlate with the swimming speed in the 25 and 35 m sprints and there was no correlation between the VLa_max_ values in the 3 sprints. This is in agreement with the assumption that there is a short period at the onset of maximal exercise during which increased lactate production has not begun [6] and justifies the subtraction of a t_alac_ from the t_exer_ in Equation (1). On the other hand, our finding that the VLa_max_ results were qualitatively the same whether based on the original t_alac_ assumptions or on the “1.5×” scenario (the one using 1.5 times the original t_alac_ values) shows that reasonable variations in the t_alac_ do not impact considerably on the value of the VLa_max_ as an index of the anaerobic lactic power. Thus, our proposed t_alac_ values appear to be valid.

A possible limitation of our study is the heterogeneity of the sample in terms of sex and stroke, which was dictated by our choice to include only highly trained and elite swimmers in the study. The different maturation levels of the participants might be another limitation. On the other hand, this may allow a generalization of our findings on the value of the VLa_max_ as an index of anaerobic lactic power. Future studies could examine blood lactate responses (La_max_, tLa_max_, and VLa_max_) in larger and/or more homogeneous samples, as well as in non-elite athletes.

## 5. Practical Applications

The following practical applications emerge from the present study:The VLa_max_ is a promising index of anaerobic lactic power that can be rather easily determined in sprints lasting about 10 to 30 s.Because the tLa_max_ was about 2 min in all the sprints, we recommend starting repeated blood sampling 1 min after the completion of maximal swimming so as not to miss the La_max_.The sprint swimming performance can be improved by designing training programs that raise the VLa_max_.

## 6. Conclusions

By examining the blood lactate response of elite swimmers to maximal efforts of 25, 35, and 50 m, the present study showed the highest anaerobic lactic power (as judged from the VLa_max_) at the shortest distance but its best correlation with performance at the longest distance. Thus, sprints lasting 10 to 15 s seem to be most appropriate for assessing the maximal VLa_max_, although sprints lasting about 30 s may have a better predictive ability of performance. The finding that tLa_max_ was approximately 2 min necessitates an early start of post-exercise blood sampling (at about 1 min) for the accurate measurement of the La_max_ and, hence, the VLa_max_. Finally, the positive correlation of the sprint swimming speed with the VLa_max_ suggests the latter as an index of anaerobic lactic power, much like VO_2_max is an index of aerobic power.

## Figures and Tables

**Table 1 sports-11-00087-t001:** Characteristics of the swimmers (mean (SD)).

	Male (*n* = 8)	Female (*n* = 6)
Age (y)	21.7 (4.8)	18.2 (3.0)
Body mass (kg)	83 (9)	61 (5)
Height (m)	1.87 (0.06)	1.72 (0.05)
Body mass index (kg·m^–2^)	23.6 (1.8)	20.8 (1.5)

**Table 2 sports-11-00087-t002:** Swimming stroke specialization and FINA points of each swimmer.

Swimmer	Sex	Swimming Stroke Specialization	FINA Points
1	M	Freestyle	821
2	M	Freestyle	703
3	M	Freestyle	721
4	M	Freestyle	782
5	M	Freestyle	873
6	M	Butterfly	751
7	M	Butterfly	628
8	M	Breaststroke	823
9	F	Freestyle	637
10	F	Freestyle	736
11	F	Backstroke	654
12	F	Backstroke	790
13	F	Backstroke	764
14	F	Backstroke	701

F, female; M, male.

**Table 3 sports-11-00087-t003:** Swimming time and speed at the 25, 35, and 50 m sprints of each swimmer.

Swimmer	Sex	Stroke	Swimming Time (s)			Swimming Speed (m·s^–1^)		
			25 m	35 m	50 m	25 m	35 m	50 m
1	M	Freestyle	10.14	15.50	23.28	2.47	2.26	2.15
2	M	Freestyle	10.34	16.11	24.22	2.42	2.17	2.06
3	M	Freestyle	10.82	16.31	24.23	2.31	2.15	2.06
4	M	Freestyle	10.14	15.37	23.29	2.47	2.28	2.15
5	M	Freestyle	10.23	14.96	22.47	2.13	2.00	1.91
6	M	Butterfly	10.93	16.71	24.79	2.29	2.09	2.02
7	M	Butterfly	11.74	17.47	26.17	2.13	2.00	1.91
8	M	Breaststroke	12.31	19.10	28.81	2.03	1.83	1.74
9	F	Freestyle	12.38	18.48	28.10	2.02	1.89	1.78
10	F	Freestyle	11.27	17.38	26.12	2.22	2.01	1.91
11	F	Backstroke	13.68	21.03	31.88	1.83	1.66	1.57
12	F	Backstroke	13.31	20.16	30.20	1.88	1.74	1.66
13	F	Backstroke	13.83	20.40	30.65	1.81	1.72	1.63
14	F	Backstroke	13.32	19.68	30.65	1.88	1.78	1.63
Mean (SD)			11.75 (1.38)	17.76 (2.04)	26.78 (3.21)	2.16 (0.25)	1.99 (0.23)	1.89 (0.22)

F, female; M, male.

**Table 4 sports-11-00087-t004:** Blood Lactate Responses to the 25, 35, and 50 m Sprints (*n* = 14, mean (SD)).

Distance (m)	La_pre_ (mmol·L^−1^) *	La_max_ (mmol·L^−1^) *	tLa_max_ (min)	VLa_max_ (mmol·L^−1^·s^−1^) ^#^
25	1.0 (0.5)	7.0 (1.5)	2.2 (0.8)	0.75 (0.18)
35	2.1 (0.6)	9.3 (2.2)	2.1 (0.7)	0.54 (0.18)
50	3.4 (1.3)	13.8 (2.6)	2.4 (1.0)	0.49 (0.16)

La_pre_, lactate before the sprint; La_max_, maximal lactate after the sprint; tLa_max_, time to reach maximal lactate after the sprint; VLa_max_, maximal lactate accumulation rate. * Significant differences between all 3 distances (*p* < 0.001). ^#^ Significant differences between 25 and 35 m, as well as between 25 and 50 m (*p* < 0.001).

## Data Availability

All study data are available upon request.

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
