# Peer review of "Blood Lactate and Maximal Lactate Accumulation Rate at Three Sprint Swimming Distances in Highly Trained and Elite Swimmers"

_sports, 2023, doi:10.3390/sports11040087_

Round 1

Reviewer 1 Report

Firstly, I would like to recognize the authors for the data they collected and the preparation of the manuscript. I believe that the most significant finding of the project is the fact that the lactate peaked approximately two minutes after each swim. This provides essential guidance as to when the lactate should be measured in order not to miss the peak level.

My initial feeling was that the sample was small and the population was rather heterogeneous. Despite the limitations of the study that are presented by the authors I still believe that the results will benefit both swimmers as well as swim coaches and researchers.

The title is clear and accurate.

The abstract presents the rationale of the study, and the general results are presented.

The introduction is clear and follows a logical sequence while all the relevant scientific literature is presented.

The materials and methods section is presented with sufficient detail so that someone can replicate and build on the published results.

The results were discussed, interpreted, and compared to previous studies in the discussion section.

Author Response

Thank you for these positive comments!

Reviewer 2 Report

The paper is potentially interesting for swimming and triathletes coaches and researchers in the field. 

However, I have some doubts on many topics of the paper.

In the introduction is cited only one study who reported value of 22 mml/L of lactate after a 15 mm swimming. This is a very weak evidence. Also, the study of Heck does not have any other supporting study confirming his findings. Using 1 study to support the main hypotheses of the paper is tto few.

The main problem as highlighted by the authors, is the small sample size. I understand is not easy to find top level athletes available to be subjects in a research, but this does not justify the small and heterogenic sample size: male and female mixed and different swimming techniques. 

A little observation when you cite water temperature. Is not cleas when it was 29 and when if was 26.5.

A light warm up is not appropriate to elicit a subsequent maximal lactate response. An appropriate warm up to elicit maximal lactate response in a following trial must be close to lactate max and have appropriate duration, as recommended by several studies  (e.g. Gerbino A, Ward SA, Whipp BJ. Effects of prior exercise on pulmonary gas-exchange kinetics during high-intensity exercise in humans. J Appl Physiol. 1996;80(1):99–107) . 

Half lactate recovery happens in the following 30 min post exercise, thus the choice of 30 minutes interval between trials is appropriate for the shortest distances, but maybe is too short for the 50m. A sound protocol, could have been to test the 50 m in a different day and to have all measures at least in duplicate (better triplicate) in order to have a more reliable measurement of LA.

Please change the abstract because the LA was not measured continuously, (it is possible only by mean of vein measurements) but every minute.  

the use of talac is very problematic because there are few supporting studies, and it is, as you stated, a theoretical determination of its length.

please provide Shapiro-Wilks results.

It is very weird to me as the statistical power could be so high. Please provide further details on its calculation.

correlation of VLa is acceptable only with the 50 m . 

A methodological observation on using "scenario". These are hypothetical data, is not scientific sound to make "scenario" without experimental data. 

The fact that different authors found different La levels depending of swimming styles, kind of subjects and level of qualification, pose a serious question on the usability of the results.  

At line 259, you mention also running, assuming the same results of swimming. It is clear since long time that running and swimming can't be compared anyway , being swimming not able to elicit the maximal lactate response (e.g. PO Åstrand & Bengt Saltin. Maximal oxygen uptake and heart rate In various types of muscular activity.J. Appl. Physiol. 16(6):977-981,1961 .

In summary, the paper is scientifically weak. I recommend a deep revision.

P

Author Response

Thank you for your constructive comments. Below please find our point-by-point responses.

Comment: The paper is potentially interesting for swimming and triathletes coaches and researchers in the field.

However, I have some doubts on many topics of the paper.

In the introduction is cited only one study who reported value of 22 mml/L of lactate after a 15 mm swimming. This is a very weak evidence. Also, the study of Heck does not have any other supporting study confirming his findings. Using 1 study to support the main hypotheses of the paper is tto few.

Response: We could not find any other study reporting blood lactate after a single swimming sprint below 50 m, and this is why we decided to study it. We would be happy to consider any relevant reference that the referee would bring to our attention. The same holds for the study of Heck et al., that is, we decided to study the matter due to the dearth of studies. We did not actually base any hypothesis on that paper. We just studied one distance based on Heck et al. (25 m) and two longer distances to either confirm or refute their findings.

Comment: The main problem as highlighted by the authors, is the small sample size. I understand is not easy to find top level athletes available to be subjects in a research, but this does not justify the small and heterogenic sample size: male and female mixed and different swimming techniques. 

Response: Whether a sample size is small or adequate is determined by the statistical power. As shown throughout the Results section, our study was sufficiently powered; thus, the sample size cannot be considered small. As for the heterogeneity of the sample, we ourselves acknowledge it as a limitation. However, we accepted this limitation in view of the “greater good,” that is, to examine these top-level athletes in their specialization stroke, where they would perform best.

Comment: A little observation when you cite water temperature. Is not cleas when it was 29 and when if was 26.5.

Response: 29 refers to air temperature, whereas 26.5 refers to water temperature. We changed the expression to make this clearer (l. 111).

Comment: A light warm up is not appropriate to elicit a subsequent maximal lactate response. An appropriate warm up to elicit maximal lactate response in a following trial must be close to lactate max and have appropriate duration, as recommended by several studies  (e.g. Gerbino A, Ward SA, Whipp BJ. Effects of prior exercise on pulmonary gas-exchange kinetics during high-intensity exercise in humans. J Appl Physiol. 1996;80(1):99–107) . 

Response: The study mentioned by the reviewer is a study in cycling that involved bouts lasting 6 minutes; these bouts elicited blood lactate responses below 6 mmol/L that did not differ between experimental conditions. Thus, we do not see how it is related to our study or how it supports the reviewer’s claim. In fact, considerable disagreement surrounds the issue of which type of warm-up is best for eliciting a subsequent maximal lactate response or maximal performance. Recommendations range from short in-water warm-ups of as little as 600 m to extensive in-water warm-ups with additional out-of-water exercises or passive heating (Neiva et al., The effects of different warm-up volumes on the 100-m swimming performance: a randomized crossover study, J Strength Cond Res 29: 3026–3036, 2015; Neiva et al., Warm-up and performance in competitive swimming, Sports Med 44:319–330, 2014; Czelusniak et al., Effects of warm-up on sprint swimming performance, rating of perceived exertion, and blood lactate concentration: A systematic review. J Funct Morphol Kinesiol 6:85, 2021). Thus, and in view of the fact that our subjects would have to perform three maximal sprints within about one hour, we decided not to burden them with an additional maximal effort during warm-up.

Comment: Half lactate recovery happens in the following 30 min post exercise, thus the choice of 30 minutes interval between trials is appropriate for the shortest distances, but maybe is too short for the 50m. A sound protocol, could have been to test the 50 m in a different day and to have all measures at least in duplicate (better triplicate) in order to have a more reliable measurement of LA.

Response: Please note that we did not perform any test after the 50-m test; we only employed 30 min of rest after the 25- and 35-m tests, which, as you point out, were sufficient. It would be great to have these top-level athletes at our disposal on multiple days, but it was impossible to fit this into their training schedule. In fact, we feel lucky that the athletes and their coaches agreed to deviate from their routine even for this one day.

Comment: Please change the abstract because the LA was not measured continuously, (it is possible only by mean of vein measurements) but every minute.  

Response: We have not used the adverb “continuously” but, rather, “continually,” meaning “frequently” or “regularly” (https://www.scribbr.com/commonly-confused-words/continually-vs-continuously/). Nevertheless, we have added “every minute” for clarity.

Comment: the use of talac is very problematic because there are few supporting studies, and it is, as you stated, a theoretical determination of its length.

Response: We agree that there are few supporting studies. However, this does not necessarily make talac “very problematic.” It just makes it worth studying further. In our study, we provide support for the notion of talac by showing its robustness against reasonable variation. We believe that such observations deserve publication, so that other researchers can test them.

Comment: please provide Shapiro-Wilks results.

Response: Done (ll. 155–156).

Comment: It is very weird to me as the statistical power could be so high. Please provide further details on its calculation.

Response: As mentioned under Statistical Analysis, statistical power was derived from SPSS during ANOVA. The fact that it was so high is due to the fact that effect sizes (eta-squared) were also very high.

Comment: correlation of VLa is acceptable only with the 50 m . 

Response: We are not sure we understand what you mean by this. If you mean it was highest at 50 m, we agree. But it was also satisfactory (that is, r > 0.5) in other distances too.

Comment: A methodological observation on using "scenario". These are hypothetical data, is not scientific sound to make "scenario" without experimental data. 

Response: By definition, a scenario is “a description of possible events” (https://dictionary.cambridge.org/dictionary/english/scenario). Thus, it can be (and usually is) hypothetical.

Comment: The fact that different authors found different La levels depending of swimming styles, kind of subjects and level of qualification, pose a serious question on the usability of the results. 

Response: Finding different lactate levels between studies is completely natural, even with the same swimming styles, kind of subjects and level of qualification, due to well-documented inter- and intra-individual variability. What was important in our study was not having uniform lactate levels but, rather, studying VLamax in athletes who shared a very high level and a maximal effort. 

Comment: At line 259, you mention also running, assuming the same results of swimming. It is clear since long time that running and swimming can't be compared anyway , being swimming not able to elicit the maximal lactate response (e.g. PO Åstrand & Bengt Saltin. Maximal oxygen uptake and heart rate In various types of muscular activity.J. Appl. Physiol. 16(6):977-981,1961 .

Response: We removed the reference to running.

Comment: In summary, the paper is scientifically weak. I recommend a deep revision.

Response: We respectfully disagree with this characterization. We believe that our paper constitutes an important contribution to the quest for an index of anaerobic lactic power and that it has greatly improved following your valuable comments.

Round 2

Reviewer 2 Report

Dear Author, I appreciate your effort, however  IMHO some doubts remains, please check it again.

In the introduction is cited only one study who reported value of 22 mml/L of lactate after a 15 mm swimming. This is a very weak evidence. Also, the study of Heck does not have any other supporting study confirming his findings. Using 1 study to support the main hypotheses of the paper is too few.

Response: We could not find any other study reporting blood lactate after a single swimming sprint below 50 m, and this is why we decided to study it. We would be happy to consider any relevant reference that the referee would bring to our attention. The same holds for the study of Heck et al., that is, we decided to study the matter due to the dearth of studies. We did not actually base any hypothesis on that paper. We just studied one distance based on Heck et al. (25 m) and two longer distances to either confirm or refute their findings.

Of course, you can’t find any study on BLA below 50 m, because in this case, the energy supply is mainly ATP-CP system, thus there was no physiological (and logical) reason to measure it.  The fact that does not exist any paper on the topic in 100 years of LA studies, must arise a doubt.  Thus, the Heck studies itself is a kind of unuseful.

Comment: The main problem as highlighted by the authors, is the small sample size. I understand is not easy to find top level athletes available to be subjects in research, but this does not justify the small and heterogenic sample size: male and female mixed and different swimming techniques.

Response: Whether a sample size is small or adequate is determined by the statistical power. As shown throughout the Results section, our study was sufficiently powered; thus, the sample size cannot be considered small. As for the heterogeneity of the sample, we ourselves acknowledge it as a limitation. However, we accepted this limitation in view of the “greater good,” that is, to examine these top-level athletes in their specialization stroke, where they would perform best.

Again, you put together males and females, that is,  physiologically absurd.  This not just justify the “greater good” because many studies exist in top level swimmers.  Also, considering different strokes, is like to compare different sports…that harms seriously the result. At least you have to split male and female results.

Comment: A little observation when you cite water temperature. Is not clear when it was 29 and when if was 26.5.

OK

Response: 29 refers to air temperature, whereas 26.5 refers to water temperature. We changed the expression to make this clearer (l. 111).

Comment: A light warm up is not appropriate to elicit a subsequent maximal lactate response. An appropriate warm up to elicit maximal lactate response in a following trial must be close to lactate max and have appropriate duration, as recommended by several studies (e.g. Gerbino A, Ward SA, Whipp BJ. Effects of prior exercise on pulmonary gas-exchange kinetics during high-intensity exercise in humans. J Appl Physiol. 1996;80(1):99–107) .

Response: The study mentioned by the reviewer is a study in cycling that involved bouts lasting 6 minutes; these bouts elicited blood lactate responses below 6 mmol/L that did not differ between experimental conditions. Thus, we do not see how it is related to our study or how it supports the reviewer’s claim. In fact, considerable disagreement surrounds the issue of which type of warm-up is best for eliciting a subsequent maximal lactate response or maximal performance. Recommendations range from short in-water warm-ups of as little as 600 m to extensive in-water warm-ups with additional out-of-water exercises or passive heating (Neiva et al., The effects of different warm-up volumes on the 100-m swimming performance: a randomized crossover study, J Strength Cond Res 29: 3026–3036, 2015; Neiva et al., Warm-up and performance in competitive swimming, Sports Med 44:319–330, 2014; Czelusniak et al., Effects of warm-up on sprint swimming performance, rating of perceived exertion, and blood lactate concentration: A systematic review. J Funct Morphol Kinesiol 6:85, 2021). Thus, and in view of the fact that our subjects would have to perform three maximal sprints within about one hour, we decided not to burden them with an additional maximal effort during warm-up.

I understand your point of view, but IMHO this arms seriously the results.  I understand is practical, but arms the result the same.

Comment: Half lactate recovery happens in the following 30 min post exercise, thus the choice of 30 minutes interval between trials is appropriate for the shortest distances, but maybe is too short for the 50m. A sound protocol, could have been to test the 50 m in a different day and to have all measures at least in duplicate (better triplicate) in order to have a more reliable measurement of LA.

Response: Please note that we did not perform any test after the 50-m test; we only employed 30 min of rest after the 25- and 35-m tests, which, as you point out, were sufficient. It would be great to have these top-level athletes at our disposal on multiple days, but it was impossible to fit this into their training schedule. In fact, we feel lucky that the athletes and their coaches agreed to deviate from their routine even for this one day.

I understand this point. 

Comment: Please change the abstract because the LA was not measured continuously, (it is possible only by mean of vein measurements) but every minute.

Response: We have not used the adverb “continuously” but, rather, “continually,” meaning “frequently” or “regularly” (https://www.scribbr.com/commonly-confused-words/continually-vs-continuously/). Nevertheless, we have added “every minute” for clarity.

This is clear now.

Comment: the use of talac is very problematic because there are few supporting studies, and it is, as you stated, a theoretical determination of its length.

Response: We agree that there are few supporting studies. However, this does not necessarily make talac “very problematic.” It just makes it worth studying further. In our study, we provide support for the notion of talac by showing its robustness against reasonable variation. We believe that such observations deserve publication, so that other researchers can test them.

This stamen does not make Talac less problematic, and in fact, is very few used in sound scientific literature.

Comment: please provide Shapiro-Wilks results.

Response: Done (ll. 155–156). Ok, now it is more understandable.

Comment: It is very weird to me as the statistical power could be so high. Please provide further details on its calculation.

Response: As mentioned under Statistical Analysis, statistical power was derived from SPSS during ANOVA. The fact that it was so high is due to the fact that effect sizes (eta-squared) were also very high.

This is an effect of the very small sample size.

Comment: correlation of VLa is acceptable only with the 50 m .

Response: We are not sure we understand what you mean by this. If you mean it was highest at 50 m, we agree. But it was also satisfactory (that is, r > 0.5) in other distances too.

Correlation around  0.5 can be considered weak. Especially with a such small sample size. For reference: http://users.sussex.ac.uk/~grahamh/RM1web/Eight%20things%20you%20need%20to%20know%20about%20interpreting%20correlations.pdf.

Comment: A methodological observation on using "scenario". These are hypothetical data, is not scientific sound to make "scenario" without experimental data.

Response: By definition, a scenario is “a description of possible events” (https://dictionary.cambridge.org/dictionary/english/scenario). Thus, it can be (and usually is) hypothetical.

Of course, but they are projection of possible facts extrapolated from real data, can be a speculation of interest but has nothing to do with measurements.

Comment: The fact that different authors found different La levels depending on swimming styles, kind of subjects and level of qualification, pose a serious question on the usability of the results.

Response: Finding different lactate levels between studies is completely natural, even with the same swimming styles, kind of subjects and level of qualification, due to well-documented inter- and intra-individual variability. What was important in our study was not having uniform lactate levels but, rather, studying VLamax in athletes who shared a very high level and a maximal effort.

Of course, but this harm the assumption of generalization of results. I understand the athletes are of high level, but must be clarify what means studying, and which are the results and the relevance for training.  Results and relevance for training must be better explained.

Comment: At line 259, you mention also running, assuming the same results of swimming. It is clear since long time that running and swimming can't be compared anyway, being swimming not able to elicit the maximal lactate response (e.g. PO Åstrand & Bengt Saltin. Maximal oxygen uptake and heart rate In various types of muscular activity.J. Appl. Physiol. 16(6):977-981,1961 .

Response: We removed the reference to running.ok

Comment: In summary, the paper is scientifically weak. I recommend a deep revision.

Response: We respectfully disagree with this characterization. We believe that our paper constitutes an important contribution to the quest for an index of anaerobic lactic power and that it has greatly improved following your valuable comments.

IMHO the paper need further revisions: first, split male and females results and consider it separately.  There is not scientific basis to consider them together.

Last, please include in the limitations all the aspects above mentioned .

Author Response

Dear reviewer,

It grieves me and my co-authors to see that, although we have adequately addressed your concerns, you insist on unsubstantiated demands that betray a smatter of the scientific method, exercise metabolism, and statistics. What is more, you do this while using derogatory words like “weird” and “absurd”, which I have never used toward an author in my 30+ years as reviewer. You also use (three times!) an undefined acronym (IMHO) that is common in internet conversation slang but inappropriate in scientific writing. As a respected senior researcher, ranking among the top 100,000 scientists worldwide in all scientific fields and top 2% in the field of Sport Sciences, I am offended by such language. Please consider the following:

  1. You are wrong in believing that, in swims below 50 m, the energy supply is mainly from the ATP-CP system. Gastin (Sports Med 31: 725–741, 2001) has long and convincingly shown, by reviewing tens of relevant studies, that the ATP-CP system is NOT the dominant energy system in maximal exercises lasting over 10 s. Instead, the lactate system is. Numerous original studies before and after him have confirmed this. Thus, the lactate system is the main energy system in the 25- and 35-, as well as the 50-m swims. Although there has been just one study in swimming in such short efforts, there have been many studies on lactate production in maximal bouts lasting 10–15 s in other sports, such as running and cycling (see refs 8, 16, 20, and 21). Hence, there IS “physiological (and logical) reason” to measure lactate in such exercise.

  1. The stated aim of our paper is “investigating the blood lactate response of highly trained and elite swimmers to maximal efforts of 25, 35, and 50 m”. It is NOT to examine differences between sexes. Separating sexes, as you insist on asking, would produce two samples that would be too small (8 men and 6 women) for adequate power and, hence, valid findings. How can you ask us to “split male and females results and consider it separately” when you consider a sample size of 14 small? The literature is replete with studies that, for a variety of reasons, have used mixed samples without separating sexes. See, for example, Sperlich et al., Eur J Appl Physiol 110: 1029–1036, 2010; Wahl et al., J Strength Cond Res 31: 3489–3496, 2017; and ref. 8 just in the area of lactate research.

  1. You cannot state (referring to the issue of warm-up), “I understand your point of view, but IMHO this arms [apparently meaning harms] seriously the results. I understand is practical, but [h]arms the result the same” without posing any argument to refute our fully documented response.

  1. You cannot dismiss the existence of a short period at the onset of exercise during which increased lactate production has not yet begun (talac). A lag in increased lactate production is quite reasonable considering that 12 consecutive reactions are needed for glycogen to produce lactate (11 for glucose). Besides, doesn’t your (erroneous) belief that the energy supply is mainly from the ATP-CP system in swims below 50 m (that is, lasting about 25 s) require a dormant lactate system during this period? Let me add here that my service as reviewer for 44 journals and editor for another 6 (including 3 MDPI journals) entitles me to the opinion that a reviewer’s role is not to stifle research that contradicts his beliefs or that has few supporting studies (after all, this is what novelty is about), but to make constructive criticism that will bring this research to a scientifically acceptable level and let it be published.

  1. You cannot arbitrarily dismiss statistically significant correlations for having an r of 0.5. In our manuscript we did not call them high or low, strong or weak, just significant.

In conclusion, we are unable to make any further changes to our manuscript without compromising its aim, validity, and message. That would change it into something that does not represent us.